# Evaluating Sorafenib (SORA-2) as Second-Line Treatment for Unresectable Hepatocellular Carcinoma: A European Retrospective Multicenter Study [note 1]

**DOI:** 10.3390/cancers17060972

**Published:** 2025-03-13

**Authors:** Christian Möhring, Moritz Berger, Farsaneh Sadeghlar, Xin Zhou, Taotao Zhou, Malte Benedikt Monin, Kateryna Shmanko, Sabrina Welland, Friedrich Sinner, Birgit Schwacha-Eipper, Ulrike Bauer, Christoph Roderburg, Angelo Pirozzi, Najib Ben Khaled, Peter Schrammen, Lorenz Balcar, Matthias Pinter, Thomas J. Ettrich, Anna Saborowski, Marie-Luise Berres, Enrico N. De Toni, Tom Lüdde, Lorenza Rimassa, Ursula Ehmer, Marino Venerito, Iuliana-Pompilia Radu, Ingo G. H. Schmidt-Wolf, Arndt Weinmann, Arndt Vogel, Matthias Schmid, Jörg C. Kalff, Christian P. Strassburg, Maria A. Gonzalez-Carmona

**Affiliations:** 1Department of Medicine I, University Hospital of Bonn, Venusberg-Campus 1, 53127 Bonn, Germany; christian.moehring@ukbonn.de (C.M.); farsaneh.sadeghlar@ukbonn.de (F.S.); xin.zhou@ukbonn.de (X.Z.); taotao.zhou@ukbonn.de (T.Z.); malte_benedikt.monin@ukbonn.de (M.B.M.); christian.strassburg@ukbonn.de (C.P.S.); 2Department IB of Internal Medicine, German Armed Forces Central Hospital, 56072 Koblenz, Germany; 3Institute for Medical Biometry, Informatics and Epidemiology, Medical Faculty, University of Bonn, 53127 Bonn, Germany; moritz.berger@imbie.uni-bonn.de (M.B.); sekretariat@imbie.uni-bonn.de (M.S.); 4Core Facility Biostatistics, Central Institute of Mental Health, Medical Faculty Mannheim, Heidelberg University, 68159 Mannheim, Germany; 5Infektionsmedizinisches Centrum Hamburg (ICH), 20146 Hamburg, Germany; 61st Department of Medicine, University Medical Center of the Johannes Gutenberg University, 55131 Mainz, Germany; kateryna.shmanko@unimedizin-mainz.de (K.S.); arndt.weinmann@unimedizin-mainz.de (A.W.); 7Department of Gastroenterology, Hepatology and Endocrinology, Hannover Medical School, 30625 Hannover, Germany; welland.sabrina@mh-hannover.de (S.W.); saborowski.anna@mh-hannover.de (A.S.); vogela@icloud.com (A.V.); 8Department of Gastroenterology, Hepatology and Infectious Diseases, Otto-Von-Guericke University Hospital, 39120 Magdeburg, Germanym.venerito@med.ovgu.de (M.V.); 9Hepatology-Department of Biomedical Research, University of Bern, 3012 Bern, Switzerland; birgit.schwacha-eipper@insel.ch (B.S.-E.); iuliana-pompilia.radu@insel.ch (I.-P.R.); 10Department of Clinical Medicine—Clinical Department for Internal Medicine II, TUM School of Medicine and Health, Technical University of Munich, 80333 Munich, Germany; ulrike.bauer@mri.tum.de (U.B.); ursula.ehmer@tum.de (U.E.); 11Clinic for Gastroenterology, Hepatology and Infectious Diseases, University Hospital Düsseldorf, 40225 Düsseldorf, Germany; christoph.roderburg@med.uni-duesseldorf.de (C.R.);; 12Medical Oncology and Hematology Unit, IRCCS Humanitas Research Hospital, 20089 Rozzano, Italy; angelo.pirozzi@cancercenter.humanitas.it (A.P.); lorenza.rimassa@hunimed.eu (L.R.); 13Department of Biomedical Sciences, Humanitas University, 20072 Pieve Emanuele, Italy; 14Department of Medicine II, University Hospital, LMU Munich, 81377 Munich, Germany; najib.benkhaled@med.uni-muenchen.de (N.B.K.); enrico.detoni@med.uni-muenchen.de (E.N.D.T.); 15Medical Department III, University Hospital of Aachen, 52074 Aachen, Germanymberres@ukaachen.de (M.-L.B.); 16Division of Gastroenterology & Hepatology, Department of Medicine III, Medical University of Vienna, 1090 Vienna, Austria; lorenz.balcar@meduniwien.ac.at (L.B.); matthias.pinter@meduniwien.ac.at (M.P.); 17Department of Internal Medicine I, University Hospital Ulm, 89081 Ulm, Germany; thomas.ettrich@uniklinik-ulm.de; 18Department of Visceral Surgery and Medicine, Inselspital, University of Bern, 3012 Bern, Switzerland; 19Department of Integrated Oncology, Center for Integrated Oncology (CIO), University Hospital of Bonn, 53127 Bonn, Germany; ingo.schmidt-wolf@ukbonn.de; 20Division of Gastroenterology and Hepatology, Toronto General Hospital, Toronto, ON M5G 2C4, Canada; 21Department of Surgery, University Hospital of Bonn, 53127 Bonn, Germany; joerg.kalff@ukbonn.de

**Keywords:** hepatocellular carcinoma, liver cancer, second-line therapy, sorafenib

## Abstract

The study provides valuable insights into the use of sorafenib as a second-line (2L) treatment for advanced hepatocellular carcinoma (HCC) in a European cohort, significantly expanding the current evidence base, which is only derived from a few retrospective studies in Asian populations. The findings are particularly relevant for clinicians managing patients with unresectable HCC, especially those with preserved liver function and low AFP levels, who may benefit most from sorafenib. These results can inform clinical decision-making and highlight the need for further research to refine treatment strategies.

## 1. Introduction

Hepatocellular carcinoma (HCC) continues to be a severe global health challenge, ranking as the sixth most common cancer type worldwide and the fourth leading cause of cancer-related deaths. HCC is often diagnosed at an advanced stage, leading to limited curative treatment options [1].

To accurately assess patients and guide treatment decisions, the Barcelona Clinic of Liver Cancer (BCLC) classification considers tumor spread, performance status, and liver function (Child–Pugh Classification) [2,3,4]. When macrovascular infiltration and/or extrahepatic metastases (BCLC C) are present, and local-interventional therapies are not indicated, or progressive disease occurs after local therapies, systemic therapy becomes the standard of care in patients with preserved liver function [2].

The landscape of systemic treatments for advanced HCC has seen considerable changes in recent years. For over a decade, the tyrosine kinase inhibitor (TKI) sorafenib was the sole effective systemic therapy option [5]. However, in 2018, the REFLECT study demonstrated non-inferiority for the TKI lenvatinib compared to sorafenib, establishing lenvatinib as an equal first-line therapy option [6]. The IMbrave150 trial, published in 2020, revealed a significant increase in survival rates with the combination of atezolizumab (Anti-PD-L1 antibody) and bevacizumab (VEGF inhibitor) (atezo/bev) compared to sorafenib, thus establishing the era of immune checkpoint inhibitors (ICIs) in first-line therapy for advanced HCC [7]. The recent HIMALAYA trial further complements the current first-line approved treatment options in Europe, showing superior survival data for the combined immune checkpoint inhibition with tremelimumab (Anti-CTLA-4 antibody) and durvalumab (Anti-PD-L1 antibody) against sorafenib [8].

Moreover, the CARES-310 study, comparing outcomes with the camrelizumab (PD-1 antibody) and rivoceranib (TKI inhibiting vascular endothelial growth factor receptor-2) against sorafenib, revealed a significantly improved median overall survival (OS) [9]. More recently, in the phase III CheckMate 9DW trial the PD-1 antibody, nivolumab, plus the anti-CTLA-4 antibody, ipilimumab, showed improved OS vs. lenvatinib or sorafenib [10]. Concurrently, the ORIENT-32 study facilitated the approval in China of the sintilimab (PD-1 antibody) and IBI305 (a bevacizumab biosimilar) combination for first-line HCC treatment, showcasing significantly longer OS compared to sorafenib [11]. Additionally, in the HIMALAYA trial, the PD-L1 antibody durvalumab and in the RATIONALE-301 trial the PD1-antibody tislelizumab as monotherapy showed noninferiority for OS against sorafenib in the first-line setting [8,12].

Thus, these studies anticipate a further increasing complexity in the landscape of first-line therapies of advanced HCC.

Following progression or discontinuation of first-line therapy, between 38% and 56% of patients are eligible for second-line (2L) systemic treatment [13,14,15]. Current 2L recommendations encompass several tyrosine kinase inhibitors or angiogenic agents, such as sorafenib, lenvatinib, regorafenib, cabozantinib or ramucirumab, but also ICIs, such as pembrolizumab or nivolumab in combination with ipilimumab [2,16,17,18,19,20,21]. Although most of these recommendations are based on data following first-line sorafenib treatment, available evidence suggests similar efficacy after ICI therapy [22,23,24]. Most recently, Chan et al. as well as El-Khoueiry et al. presented the first prospective phase II trials for cabozantinib or regorafenib in combination with prembrolizumab after ICIs [25,26].

The expanding array of first-line treatment options makes the decision-making process for 2L therapy even more complex and is further complicated by approval restrictions.

Nevertheless, the use of sorafenib or lenvatinib as 2L treatment lacks prospective data, making the choice of 2L therapy predominantly reliant on clinicians’ discretion and approval status rather than clear evidence in the era of ICIs or lenvatinib first-line treatment.

Currently, sorafenib remains one of the most frequently applied 2L therapies and the only medication that can be administered in any treatment line according to EMA approval. However, prospective survival data are only available in the first-line setting, with a median OS of 10.4–15.5 months across a multiplicity of published prospective studies [6,8,21,27]. Until now, only retrospective trials from Asia–Pacific countries and a few trials from Europe and the US included subgroups of patients treated with sorafenib as 2L treatments focusing on the analysis of effectiveness of sorafenib in this setting. For instance, a retrospective trial from Korea showed a median OS of 8.7 months for a subgroup of sorafenib (*n* = 52) after failed first-line treatment under lenvatinib [28]. Another retrospective analysis, including 29 patients treated with sorafenib after atezo/bev, also from Asia, reported a median OS of 11.2 months (95% CI: 2.7–19.6) [22]. The current strongest evidence of 2L treatment with sorafenib is the recently published retrospective analysis of Lee et al. who reported the outcome of 339 patients in Asia–Pacific countries receiving sorafenib after atezo/bev with a median OS of 6.3 months [29]. The ARTE study group from Italy reported a median OS of 6.9 months for sorafenib 2L after atezo/bev in their prospective cohort analysis (*n* = 40) [30].

Therefore, the aim of our study was to analyze the effectiveness and safety of sorafenib as a 2L therapy in a large European cohort of patients with advanced HCC after first-line therapy failure in a multicentric retrospective study.

## 2. Materials and Methods

### 2.1. Study Population

For the present retrospective multicenter study, patients with advanced HCC who underwent 2L therapy with sorafenib (SORA-2), regardless of their preceding systemic first-line treatment regimen, were evaluated for inclusion. The study duration spanned from January 2015 to May 2023. Confirmation of HCC diagnosis was requisite for enrollment, achieved either through histological or radiological means.

In total, 81 patients from 12 European tertiary centers, including 9 German, 1 Swiss, 1 Austrian, and 1 Italian centers, were enrolled in this study. To uphold the integrity of the patient pool and prevent the introduction of undue selection bias, no further inclusion/exclusion criteria were applied during the patient selection process.

The choice of a systemic 2L therapy was made by the treating physician in consultation with the patients and after exclusion of possible surgical or local therapies in interdisciplinary tumor boards. The reasons for opting for 2L therapy included disease progression under the first-line therapy or toxicity associated with the first-line therapy.

Informed consent for therapy with sorafenib was obtained from every patient. The ethical approval for the retrospective analysis was obtained by the Ethical Committees of each participating institution. The approval includes the enrollment of every patient who has given their consent in written or verbal form. The Ethics Committee of the Medical Faculty at the University of Bonn approved the study (Approval No. 341/17).

The study was conducted in accordance with the Declarations of Helsinki and Istanbul.

### 2.2. Study Design

Clinical, laboratory, and tumor-specific baseline characteristics were collected at diagnosis, before starting and upon completing sorafenib therapy. Comprehensive clinical and therapeutic data during 2L sorafenib treatment, prior systemic or local therapies, and subsequent interventions were systematically compiled.

Therapeutic response and tumor staging were evaluated by local radiology assessment using computed tomography (CT) or magnetic resonance imaging (MRI) according to the Response Evaluation Criteria in Solid Tumors (RECIST) 1.1 criteria [31]. Response to therapy was defined by the overall response rate (ORR) summing up patients who showed complete (CR) or partial response (PR) to therapy. The disease control rate (DCR) encompassed patients who had CR, PR, or stable disease (SD).

Occurrence of adverse events of graduation ≥ 3 during sorafenib administration was documented in accordance with the Common Terminology Criteria of Adverse Events (CTCAE) version 5.0 [31].

The primary endpoint of the study was OS subsequent to the start of sorafenib therapy. Secondary endpoints comprised progression-free survival (PFS) post initiation of sorafenib treatment, objective response under sorafenib, and tolerability of sorafenib therapy. Through univariable and multivariable analyses, the impact of prior therapies, tumor- and patient-specific characteristics, and liver function on OS during sorafenib treatment was additionally assessed.

OS was defined as the time interval between the initiation of sorafenib therapy and the occurrence of death, or the date of the latest follow-up. Analogously, PFS was calculated as the time interval between the initiation of sorafenib therapy and the onset of tumor progression, death, or the date of the latest follow-up.

### 2.3. Therapy Scheme

The standard dosage of sorafenib administered to patients was 800 mg per day distributed in two doses. The initial dosage was adjusted based on an assessment of toxicity risk, taking into consideration hepatic function, overall patient condition, and comorbidities. In the case of dose reduction at therapy start, the sorafenib dosage was escalated when no relevant toxicities occurred. Sorafenib administration continued until tumor progression or intolerable adverse events occurred. Dose adjustments, including reductions or interruptions, were made in response to toxicity or upon patient request.

After the initiation of sorafenib, patients were regularly examined clinically and through laboratory tests every 1–4 weeks to assess tolerability. The first radiological assessment was scheduled at 12 weeks after the initiation of sorafenib therapy. Subsequent imaging evaluations were performed every 12 weeks as standard practice. In cases where clinical suspicion or laboratory results suggested disease progression, staging examinations may be conducted ahead of the 12-week schedule to ensure timely intervention. If disease progression was confirmed, and the performance status of the patient permits, a third-line therapy was considered or re-evaluating the suitability of locoregional treatments was discussed again in multidisciplinary tumor boards, aiming to optimize patient management and outcomes.

### 2.4. Statistical Analysis

Statistical analysis was carried out using R version 4.2.3 (R Core Team 2023: R: A Language and Environment for Statistical Computing, R Foundation for Statistical Computing, Vienna, Austria). Descriptive analyses included the calculation of medians and interquartile ranges for continuous variables and frequencies (absolute and relative) for categorical variables. OS was compared with respect to tumor- and patient-specific characteristics using Kaplan–Meier curves (deriving estimates and 95% confidence intervals of median survival times), log-rank tests and univariable Cox proportional hazards regression models. Hazard ratios and corresponding 95% confidence intervals were graphically illustrated by forest plots. PFS was analyzed for the entire cohort by the Kaplan–Meier method, estimating median survival and corresponding 95% confidence intervals. To construct confidence intervals for median survival times, we used a normal approximation based on the cumulative hazard or log(survival). Complementary, a multivariable analysis of overall survival was performed including serum alpha-fetoprotein (AFP) levels and Child–Pugh class and performance status (ECOG) before the initiation of sorafenib therapy, using a survival tree with conditional inference permutation tests [32]. For this, we applied the function ctree() in the R add-on package party with significance level α = 0.1 (and the default values of the other hyperparameters) for tree size tuning.

## 3. Results

### 3.1. Patient and Therapy Characteristics

A total of 81 patients receiving sorafenib as 2L therapy from 12 centers were enrolled in the study. Baseline and therapy characteristics of all patients are presented in Table 1. The median age of patients at the initiation of sorafenib therapy was 67.8 years with 79.0% of patients being male.

The majority of patients exhibited a satisfactory performance status (ECOG 0–1: 77.7%, ECOG 2: 19.8%), and maintained liver function was assessed by ALBI grade: 27.2% ALBI 1, 48.1% ALBI 2, and 13.6% ALBI 3; and by Child–Pugh class (CP): 67.9% CP A and 18.5% CP B. Eight (9.9%) of CP B patients had a CP score of 7, three (3.7%) had a score of 8, and four (4.9%) had a score of 9, respectively.

Etiologically, alcoholic steatohepatitis (ASH) accounted for the most prevalent risk factor (34.6%), followed by hepatitis C (22.2%) and metabolic dysfunction-associated steatotic liver disease/metabolic-associated steatohepatitis (MASLD/MASH) (16.0%). Hepatitis B was diagnosed in seven (8.6%) patients. More than one underlying condition contributing to HCC was present in 21.0% of patients.

In a first-line therapy setting, 35 (43.2%) patients received atezo/bev, 36 (44.4%) patients received lenvatinib, and nine (11.1%) patients participated in a clinical trial. All patients treated within a clinical trial received ICI therapy. Furthermore, 33.3% of patients had undergone prior surgical resection and 71.6% patients had received prior local ablative therapies: 43.2% transarterial chemoembolization, 13.6% microwave ablation (MWA), 7.4% radiofrequency ablation (RFA), and 7.4% transarterial radioembolization (TARE).

Eighteen patients initiated sorafenib therapy at BCLC stage B (22.2%) and 57 patients at stage BCLC C (70.4%). Moreover, 44.4% had macrovascular invasion, with overall 30.9% exhibiting portal vein infiltration. Serum AFP levels exceeded 200 ng/mL in 35 patients (43.2%) and exceeded 400 ng/mL in 29 patients (35.8%) at the onset of therapy.

Table 2 summarizes the therapy characteristics. Only 39 (48.1%) patients received the full sorafenib dose and the median duration of therapy was 2.56 months (0.33–39.20). The primary reasons for discontinuation of sorafenib therapy were tumor progression (32 patients, 39.5%) and therapy-related toxicity (23 patients, 28.4%).

Of note, 41 (50.6%) patients received subsequent therapies following cessation of sorafenib treatment. The most common forms of subsequent therapy were systemic treatments, with 38 (46.9%) receiving one subsequent therapy, 11 (13.6%) receiving two subsequent therapies, and five (6.2%) receiving three subsequent therapies. Third-line therapies included cabozantinib in 26 cases (65.0%), atezo/bev in two cases (5.0%), nivolumab/ipilimumab in two cases (5.0%), regorafenib in four cases (10%), ramucirumab in two cases (5.0%), and lenvatinib in two cases (5.0%).

### 3.2. Effectiveness

The median follow-up duration amounted to 6.67 months (range: 1 day–31.5 months). Fifty-seven (70.4%) patients were deceased and eighteen (22.2%) were alive at the time of data collection completion. A total of six patients were lost to follow-up. The median OS for the entire cohort was 7.43 months (95% CI: 6.64–13.60), and PFS was 3.75 months (95% CI: 3.02–4.86) (shown in Figure 1).

Radiological treatment response was assessable in 55 patients (67.9%) and response rates were calculated in relation to the available patients’ data (Table 2). Three patients (5.5%) showed a PR, 20 (36.4%) patients experienced an SD, and 32 (58.2%) showed a PD. This resulted in a DCR of 41.8% and an ORR of 5.5%.

The impact of sorafenib treatment on AFP levels was also investigated. The median AFP level before the initiation of sorafenib therapy was 153 ng/mL (range: 1.7–66,600). A reduction in AFP levels by more than 20% was considered a treatment response, resulting in an AFP response rate of 25% (available data in 60 patients with 15 patients showing AFP therapy response). Patients with an AFP response also demonstrated a trend toward improved survival compared to patients who did not respond according to AFP (median OS: 14.52 months, 95% CI: 7.43—not estimable, vs. 7.13 months, 6.6–12.5; *p* = 0.08).

### 3.3. Prognostic Factors and Subgroup Analysis

The results on OS with regard to patient-specific, tumor-specific, and liver function characteristics are displayed in Figure 2, Figure 3 and Figure 4. Regarding performance status, patients with an ECOG of 0 at therapy initiation exhibited a significantly longer median OS of 15.01 months (95% CI: 9.72–20.80) compared to patients with ECOG scores of 1 (HR: 1.663, 95% CI: 0.886–3.119; 7.13 months, 95% CI: 6.47–13.60) or 2 (HR 3.023, 95% CI: 1.380–6.621; 6.60 months, 95% CI: 4.34—not estimable) (*p* = 0.017, shown in Figure 2a). Liver function also had prognostic relevance for OS in patients receiving sorafenib with CP showing a statistically significant impact (HR 2.484, 95% CI: 1.268–4.866). Patients with CP A reached a median OS of 9.79 months (95% CI: 7.13–16.2) vs. 5.78 months (95% CI: 5.16—not estimable) in patients with CP B cirrhosis (*p* = 0.006, shown in Figure 2b). Analysis of ALBI grade demonstrated a trend to better prognosis for ALBI 1 with median OS 13.60 months (95% CI: 7.43—not estimable) vs. 9.56 months (95% CI: 6.67–16.40) for ALBI 2: HR 1.120, 95% CI: 0.582–2.154; vs. 6.60 months, 95% CI: 4.83—not estimable for ALBI 3: HR 2.297, 95% CI: 0.977–5.401 (*p* = 0.120). Additionally, the BCLC stage at the start of sorafenib also had a statistically significant influence on OS, with a median OS of 16.36 months (95% CI: 9.72—not estimable) for patients in BCLC stage B and 7.03 months (95% CI: 6.60–12.50) for patients in BCLC stage C (HR: 2.296, 95% CI: 1.145–4.604; *p* = 0.016, shown in Figure 2c). Furthermore, besides the AFP therapy response, the AFP level at the initiation of sorafenib therapy also had prognostic value for OS: patients with an AFP level < 400 ng/mL had a significantly longer median OS of 13.60 months (95% CI: 7.43–16.53) compared to those with a level ≥ 400 ng/mL, who had a median OS of 6.64 months (95% CI: 5.78–9.07) (HR: 2.201, 95% CI: 1.244–3.894; *p* = 0.006, shown in Figure 2d). Moreover, OS varied significantly depending on the response to sorafenib therapy: patients with disease control had a median OS of 16.2 months (95% CI: 13.6—not estimable), while patients with progressive disease had a median OS of 7.13 months (95% CI: 5.78–13.9) (*p* = 0.003).

Interestingly, the first-line therapy (atezo/bev: median OS: 7.13 months, 95% CI: 6.60–15.0 vs. lenvatinib: 7.43 months, 95% CI: 6.47–14.9, *p* = 0.860, shown in Figure 3a) and the etiology of HCC (viral: 6.67 months, 95% CI: 5.32—not estimable vs. non-viral: 7.69 months, 95% CI: 6.67–13.60, *p* = 0.890, shown in Figure 3b) showed no prognostic relevance for 2L OS following the initiation of sorafenib. In particular, patients with ASH experienced a median OS of 11.10 months (95% CI: 7.13—not estimable), with MASLD/MASH 9.07 months (95% CI: 5.42—not estimable), with hepatitis B 13.9 months (95% CI: 5.32—not estimable), and with hepatitis C 5.32 months (95% CI: 2.14—not estimable) (*p* = 0.690).

Patients who received subsequent therapies after sorafenib, regardless of the type of tumor-specific therapy, exhibited significantly prolonged survival compared to those who received the best supportive care alone (13.6 months, 95% CI: 8.25–20.8, vs. 6.67 months, 4.83–14.9; *p* = 0.015, shown in Figure 3c).

Receiving 100% of the planned sorafenib dose did not result in a favorable OS (HR 0.793 (95% CI 0.461–1.364, shown in Figure 3d) in our cohort of patients. Furthermore, the reason for discontinuation of sorafenib also did not seem to affect the survival outcome (9.79 months for termination because of disease progress vs. 9.72 months for termination because of toxicity (*p* = 0.910)). Figure 4 summarizes the results of the univariable analysis in a forest plot.

### 3.4. Multivariable Analysis

Through multivariable analysis we identified the combined CP class and the AFP level before the start of sorafenib as significant predictors for OS. Patients with CP class B had a median OS of 5.78 months (95% CI: 5.16—not estimable), while patients with CP class A showed a median OS of 7.13 months (95% CI: 6.47—not estimable) when the AFP level before starting sorafenib was above 400 ng/mL, and even 15.51 months (95% CI: 9.79–20.89) with AFP below 400 ng/mL (as shown in Figure 5).

### 3.5. Toxicity

Adverse events with a grade ≥ 3 were evaluated in 69 patients, with no documentation available for adverse events in 12 patients. At least one adverse event with a grade ≥ 3 was reported in 41 out of 69 patients (59.4%). The most common grade ≥ 3 adverse events included hand-foot syndrome in 10 patients (14.4%), liver dysfunction in nine patients (13.0%), and diarrhea in eight patients (11.6%).

The complete description of grade ≥ 3 adverse events is presented in Table 3.

Before the initiation of therapy, 14 patients (17.2%) exhibited ascites of varying degrees, and two patients (2.4%) experienced hepatic encephalopathy. However, even though data on this aspect were available for only 63 patients following the completion of sorafenib therapy, there were 24 patients (29.6%) with ascites of any severity and 11 patients (13.5%) with some form of hepatic encephalopathy. These changes are also reflected in the CP scores before and after the initiation of sorafenib treatment. The trends in CP scores are shown in Figure 6, which indicates a tendency towards deterioration in the majority of patients.

## 4. Discussion

As checkpoint inhibitors were established as the first-line standard of care, sorafenib, once the first-line treatment, became the treatment of choice in the second-line setting [5,6,7,8]. This pragmatic approach was based on the rationale that since the mechanisms underlying the action of the TKI are distinct from those of ICIs, treatment with sorafenib would be beneficial even after progression on ICIs. However, this rationale was not supported by clinical evidence [30].

Several of the substances currently available for the treatment of advanced HCC—including cabozantinib, ramucirumab, regorafenib, pembrolizumab, or combined nivolumab/ipilimumab—were approved as second-line treatment following prior first-line treatment with sorafenib [16,17,18]. Recently, even the first prospective phase II data on 2L cabozantinib and regorafenib plus pembrolizumab after ICIs were published [25,26].

The analysis incorporates data from 81 patients with advanced HCC from 12 European tertiary centers and reveals a median OS under sorafenib as 2L treatment of 7.43 months (95% CI: 6.64–13.60) and a median PFS of 3.75 months (95% CI: 3.02–4.86). DCR was achieved in 41.8% of patients, with an ORR of 5.5%.

Comparing these results with the efficacy of sorafenib in the first-line from prospective trials, sorafenib appears to have quite similar effects in terms of therapy response and PFS. In the registration SHARP trial, the ORR was 2%, the DCR was 43%, and the median PFS was slightly longer with 5.5 months [5]. In other prospective trials, patients within the sorafenib control groups exhibited slightly higher ORR for sorafenib, ranging from 5.1% to 11% [6,8,27]. In a real-world setting, the Korean cohort of the large prospective observational GIDEON study rather resembles our data with an ORR of 2.7% and a median PFS of 2.5 months [33].

As expected for a 2L therapy, the OS achieved with sorafenib in all prospective first-line studies, including the GIDEON trial, was longer, ranging from 10.7 to 15.5 months, compared to the 7.43 months achieved with sorafenib as a 2L treatment in our retrospective observation [5,6,8,27,34,35].

While previous retrospective studies, particularly from Asia–Pacific countries, have provided important insights into the use of sorafenib in the second-line setting for advanced HCC, data on Caucasian patients remain limited. Our multicenter retrospective analysis contributes to this body of evidence by examining a large unselected European cohort, thereby complementing existing findings and helping to refine our understanding of sorafenib’s efficacy and safety in different patient populations. Although prospective data are still lacking, our study offers additional context that may aid clinicians in making informed treatment decisions, particularly in European settings.

A first retrospective study by Yoo et al. analyzed the 2L OS for sorafenib after progression on atezo/bev in 29 patients who demonstrated a promising OS of 11.2 months [22]. Another trial by Kim et al. retrospectively investigated a cohort of South Korean patients after progression under first-line treatment with lenvatinib, subsequently treated with sorafenib or nivolumab. Patients in the sorafenib group (*n* = 52), similar to our cohort, had a median OS of 8.7 months (95% CI: 3.0–14.4) and a median PFS of 3.3 months (95% CI: 2.5–4.0) [28]. Of note, the sorafenib cohort had a higher frequency of hepatitis B infections (73.1% vs. 9.9%) but similar ECOG status and liver function. In another Asian retrospective analysis including 29 patients, sorafenib after atezo/bev showed a longer median OS of 11.2 months (95% CI: 2.7–19.6) and a similar median PFS of 2.5 months [22]. The analyzed cohort exhibited preserved liver function determined by CP (100% CP A) and good performance status (100% ECOG 0–1), partially explaining the better survival data. A more recently published Asian-Pacific multicenter study reported a poorer median OS for sorafenib treatment (*n* = 339) after atezo/bev of 6.3 months (95% CI: 5.3, 7.8) and a median PFS of 2.3 (95% CI: 2.0, 2.6) [29]. Another propensity score matched analysis from South Korea reported a median OS of 7.5 months for sorafenib 2L likewise after atezo/bev [36]. The first data on sorafenib as a second-line treatment in Caucasian patients were provided by a research group from Italy. In their prospective cohort analysis, they evaluated the median OS of 40 patients, which was 6.9 months (95% CI: 2.7–11.1) [30]. A recently published retrospective multicenter study conducted across three continents also analyzed OS under sorafenib following atezolizumab/bevacizumab and reported a median OS of 7.0 months [37]. In particular, these two recent studies appear to report OS outcomes similar to those observed in our cohort.

Even considering the biases inherent in comparing different studies, our presented data, when juxtaposed with the results from randomized controlled prospective positive trials of established 2L therapies with other TKIs or anti-angiogenic antibodies, showed similar outcomes in terms of PFS or ORR. Regarding OS, our analysis initially suggests somewhat poorer outcomes when using sorafenib as 2L treatment. However, these findings should be interpreted with caution. In the prospective phase III RESORCE trial, patients progressing under sorafenib and treated with regorafenib had a longer median OS of 10.6 months (95% CI: 9.1–12.1) but similar median PFS of 3.1 months (95% CI: 2.8–4.2) and ORR of 11% [16]. However, patients in the RESORCE trial were strongly selected, including only those with tolerability to sorafenib, excluding those who discontinued sorafenib due to toxicity and permitted only patients with CP A. However, the initial approval was expanded due to the results of the REFINE trial that showed effectiveness of regorafenib regardless of prior sorafenib tolerability [38]. Recently, a prospective phase II trial showed modest efficacy in combination with pembrolizumab as 2L therapy after ICIs. The primary endpoint was the ORR. The patients were divided into two groups: first-line treatment with atezo/bev or another immune checkpoint inhibitor treatment. The ORR was 5.9% in the first group and 11.1% in the second group [26].

In the CELESTIAL trial, patients treated with cabozantinib after initial sorafenib therapy achieved a higher median OS of 11.3 months and a slightly higher median PFS of 5.5 with an ORR of 4% [17]. The better median survival could be attributed to preserved liver function (only CP A were included) and better ECOG status (only ECOG 0–1). Our cohort had 18.5% of patients with CP B and 19.8% of patients with ECOG 2. However, within the CELESTIAL trial, 28% of patients received cabozantinib as a systemic third-line treatment and 91% of patients had a BCLC C stage HCC, while our cohort had a BCLC C frequency of 70.4%, making the comparability even more complicated.

In the REACH-2 study, ramucirumab patients experienced a similar median OS of 8.5 months (95% CI: 7.0–10.6) and a median PFS of 2.8 months (95% CI: 2.8–4.1) with an ORR of 5% [18]. However, this trial investigated the efficacy of ramucirumab only in a selected patient subgroup with AFP levels > 400 ng/mL after progression under sorafenib. Our subgroup with AFP level > 400 ng/mL had a shorter median OS of 6.64 months that once again might be explained by the worse liver function and performance status (REACH-2: only CP A and ECOG 0–1).

Thus, in the interpretation of the above-mentioned prospective studies on 2L treatments in advanced HCC, particular attention must be paid to the preserved liver function of the respective cohorts. When comparing the appropriately matched subgroup of our cohort with a CP A, the OS of 9.79 months (95% CI: 7.13–16.2) suggests comparable effects of sorafenib as a 2L treatment with other approved TKIs in this therapy setting. In this context, however, the data from the prospective phase II study on therapy with cabozantinib after ICI therapy are particularly intriguing, where a median OS of 14.3 months was reported for patients with a CP A score receiving 2L cabozantinib [25].

In addition, there is also prospective data evaluating IO-based therapies in the 2L setting. Both the KEYNOTE-224 study and the randomized controlled KEYNOTE-240 study examined the use of pembrolizumab after first-line treatment with sorafenib [19,20,39,40]. Particularly noteworthy are the longer median OS results of 13.2 months (95% CI: 9.7–15.3) and 13.9 months (95% CI: 11.6–16.0), respectively, compared to those seen with TKIs or sorafenib in our cohort with a median OS of 7.43 months. Interestingly, despite the significantly longer OS, the median PFS results of 4.9 months (95% CI: 3.5–6.7) and 3.8 months (95% CI: 2.8–4.1), respectively, are comparable to the PFS in our study of 3.75 months. Both trials also showed an improved ORR of 17–18.3%. The CheckMate-040 study, a three-arm randomized controlled trial, investigated the effect of nivolumab and ipilimumab in three different dosages. The patient group treated with nivolumab 1 mg/kg plus ipilimumab 3 mg/kg every 3 weeks (4 doses) showed a remarkable median OS of 22.8 months (95% CI: 9.4—not estimable). This dual checkpoint inhibition also showed a higher ORR of 32% [21].

Similarly to sorafenib, there are no prospective data on the use of lenvatinib in the 2L setting. Several retrospective analyses from different regions (including Asia, Europe, and North America) suggest that lenvatinib may outperform sorafenib in terms of both PFS and OS by approximately 2–3 additional months, especially in relatively fit patients (Child–Pugh A, low AFP). In the available retrospective works, lenvatinib was used either after first-line treatment with sorafenib or IO-based therapy [41,42]. For instance, Qin et al. reported similar efficacy of lenvatinib in terms of OS in their Chinese cohort from three tertiary centers (*n* = 50) with a median OS of 8.5 months (95% CI: 7.5, 10.5) but longer median PFS of 5.0 (95% CI: 4.5, 6.5) months and higher ORR of 18%. The relatively promising PFS was also confirmed in two additional studies, both of which reported a significantly longer PFS with lenvatinib compared to sorafenib in the 2L setting (3.5 vs. 1.8 months and 4.8 vs. 3.3 months, respectively) [36,43]. Similarly, Chen et al. showed in their single-center study (*n* = 20) a median OS of 8.1 months (95% CI: not reported) but a slightly shorter median PFS of 3.1 months (95% CI: not reported) and again a higher ORR of 27.5% for the 2L group. Within the large multicenter trial by Lee et al., these results were confirmed with an OS of 8.0 (95% CI: 7.0, 10.9) and a PFS of 4.0 (95% CI: 3.5, 4.9) [29]. The predominantly Asian origin of these studies, and the resulting cohort differences, such as a higher prevalence of hepatitis B-related HCC, younger age distribution, and varying comorbidity profiles, may limit direct comparability to our predominantly Western-based population. However, lenvatinib as a 2L treatment appears to provide similar survival outcomes to sorafenib, although Lee et al. reported superior OS and PFS for lenvatinib compared to sorafenib within their cohort [29].

Nonetheless, these improvements with lenvatinib remain modest in some series, and second-line outcomes still vary widely depending on baseline factors such as liver function, performance status, and tumor burden. Taken together with our data, this evidence underscores that sorafenib remains a viable option in selected patients—particularly those with preserved liver function—despite the potential superiority of lenvatinib in certain subgroups. Ultimately, treatment decisions may hinge on regional approval status, local practice patterns, and individualized patient considerations.

When analyzing subgroups of patients, we found patients who particularly seem to benefit from 2L sorafenib treatment. In particular, patients with preserved liver function (Child–Pugh A) and simultaneous AFP levels < 400 ng/mL exhibited a median OS of 15.51 months (95% CI: 9.79–20.83), doubling the median OS compared to the overall cohort. It should be noted that our survival tree analysis was exploratory, identifying potential prognostic subgroups but not suggesting that any subgroup should be withheld from therapy; rather, these findings merit larger prospective validation before guiding definitive treatment decisions. In line with our findings, a meta-analysis on first-line sorafenib therapy also demonstrated improved survival for patients with Child–Pugh A [44]. Additionally, first-line sorafenib studies suggest that sorafenib can be used in patients with CP B cirrhosis. Interestingly, median OS observed in our 2L sorafenib cohort for CP B patients (5.78 months, 95% CI 5.16—not estimable) were similar to those observed in the GIDEON trial (5.2 months, 95% CI: 4.6–6.3) and the trial of Pressiani et al. (3.8 months, 95% CI: 0.4–27.3), suggesting that sorafenib therapy in the 2L setting appears to achieve similar OS outcomes to those in first-line treatment for patients with Child–Pugh B liver function [35,45].

Bruix et al. analyzed two phase III trials concerning prognostic factors and identified several predictors of first-line sorafenib OS, including macrovascular invasion, AFP levels, neutrophil-to-leukocyte ratio (NLR), extrahepatic spread, and hepatitis C [46]. Our 2L sorafenib data confirmed AFP levels ≥ 400 ng/mL (HR 2.201, 95% CI: 1.244–3.894) and extrahepatic spread (HR 1.865, 95% CI: 1.064–3.270) as prognostically relevant but failed to confirm cirrhosis etiology or any other above-mentioned factors. Interestingly, the first-line regimen did not influence the sorafenib-specific OS (HR 0.949, 95% CI: 0.534–1.685). The relatively high proportion of lenvatinib in the first-line setting is explained by the study period, which extends to a time when IO-based therapies were not yet available.

For other 2L therapies, different prognostic factors were highlighted: patients benefited most from cabozantinib treatment if they had extrahepatic spread, were treated outside of Asia, or had hepatitis B or a non-viral etiology of HCC [17]. In the RESORCE trial, patients in all subgroups equally benefited from regorafenib, and under 2L therapy with ramucirumab, ECOG performance status, presence of macrovascular invasion, and baseline AFP were significantly prognostic for OS in multivariable Cox analyses [16,18]. In retrospective analyses of lenvatinib use in 2L therapy, factors such as AFP < 400 ng/mL, no extrahepatic spread, Child–Pugh A, tumor number < 3, ORR under first-line treatment, and a PFS ≥ 6 months were positive prognostic factors for improved survival [41,42].

In addition to traditional markers such as baseline AFP and the neutrophil-to-lymphocyte ratio, recent efforts have focused on composite prognostic models to enhance the prediction of treatment response in HCC. For instance, the CRAFITY score, developed and validated by Scheiner et al., combines baseline C-reactive protein and AFP levels to stratify patients into distinct prognostic groups. This score was associated with overall survival and radiological response in patients treated with PD-(L)1-based immunotherapy, and interestingly, it also correlated with survival outcomes in a cohort of sorafenib-treated patients [47]. Given the inherent heterogeneity of HCC and the current lack of large-scale prospective studies, no single biomarker has yet emerged as a definitive predictor of sorafenib response. Future studies integrating such composite scores may help refine patient selection and guide individualized therapeutic strategies in both first- and second-line settings.

Recent literature has explored various sorafenib-based combinations in advanced HCC. For instance, a phase II trial combining sorafenib and doxorubicin in a second-line setting did not demonstrate a relevant improvement in survival, whereas the SHELTER study found an acceptable safety profile and early signs of efficacy with resminostat plus sorafenib [48,49]. More recently, a first-line trial of sorafenib and tislelizumab reported encouraging response rates and manageable toxicity [50]. However, definitive confirmation of the role of such combinations will require larger phase III studies specifically designed for second-line therapy.

Therapy tolerance and quality of life play a crucial role in palliative systemic 2L treatment. In contrast to the landmark SHARP trial, we observed a higher incidence of grade ≥ 3 toxicities in our cohort, particularly hepatotoxicity and hand-foot syndrome. This discrepancy may reflect a different patient population in 2L, often presenting with poorer liver function (Child–Pugh B) and more advanced disease at baseline. Indeed, real-world registries, such as GIDEON, have noted similarly elevated toxicity rates when sorafenib is administered to patients with higher Child–Pugh scores. Although no prospective study has directly compared sorafenib toxicity in first- vs. second-line usage, these observations suggest that the reduced hepatic reserve and overall condition typical of 2L HCC patients can lead to increased treatment-related adverse events, potentially explaining our findings. The most common adverse events with a CTCAE grade ≥ 3 in our cohort were slightly more frequent than in the SHARP study: hand-foot skin reaction (14.4% vs. 8% in SHARP), diarrhea (11.6% vs. 8% in SHARP), and fatigue (7.2% vs. 4% in SHARP). However, an especially considerable difference in the frequency of liver dysfunction was observed, occurring in 13% of our group compared to <1% in the first-line therapy trial. Furthermore, an overall of 59% of patients in our cohort experienced grade ≥ 3 toxicities, somewhat more frequent than in the SHARP study (45%) [5]. The poorer tolerability of sorafenib in 2L may be attributed to the increased number of Child–Pugh B patients in our cohort. The trends towards deterioration in CP scores observed in our cohort likely reflect the natural progression of advanced HCC, compounded by the liver’s declining capacity to tolerate systemic therapies such as sorafenib. Additionally, the high proportion of patients with Child–Pugh B at baseline may also contribute to this observed trend, as these patients are more susceptible to rapid clinical deterioration, even with close monitoring and treatment adjustments. However, these patients reflect real-world treatment scenarios, where patients with CP B also have an urgent need for therapy and are therefore treated despite guideline recommendations.

The observed deterioration in liver function, as indicated by the trends in CP scores, could be attributed to the dual impact of sorafenib. While its VEGF inhibition mechanism contributes to its anti-tumor effects, it may also lead to reduced hepatic blood flow, exacerbating hypoxia in already compromised liver cells and thus contributing to hepatotoxicity.

The elevated rate of toxicities becomes even more striking considering the shorter median treatment duration (2.56 months, range 0.33–39.20) compared to the SHARP trial (5.3 months, range 0.2–16.1), and the lower percentage of patients receiving the planned daily dose in our cohort (48.1% vs. 76% in SHARP). Nevertheless, unlike the study by Llovet et al., therapy in our cohort was more frequently terminated due to tumor progression (39.5%) and only in 28.4% due to toxicities, whereas in the SHARP trial, 29% discontinued therapy due to adverse events and 20.5% due to tumor progression [5]. In consideration of these data, one might assume that the poor tolerability of sorafenib in 2L is the reason of the short treatment duration and low treatment dosage, potentially resulting in reduced effectiveness.

The identification of the optimal therapy sequence within the relevantly expanded treatment landscape of systemic and local therapies is a crucial challenge in managing advanced HCC. Factors to consider include tumor characteristics, liver function, the presence of comorbidities, and frailty. With the markedly improved effectiveness of systemic therapy options, the multimodal therapy sequence depends on treatment response and tolerability of first-line treatment. In a large retrospective analysis, Persano et al. examined OS from the initiation of first-line therapy with either atezo/bev or lenvatinib, depending on subsequent treatments administered after PD during first-line therapy. For systemic 2L therapy following lenvatinib, there were no significant differences in OS (sorafenib HR: 1; immunotherapy HR: 0.69, 95% CI: 0.45–1.05; other therapies HR: 0.85, 95% CI: 0.52–1.38; *p* = 0.27), although the best survival was observed for patients receiving immunotherapy. Conversely, lenvatinib significantly demonstrated the best OS following atezo/bev (sorafenib HR: 1; lenvatinib HR: 0.50, 95% CI: 0.27–0.90; cabozantinib HR: 1.29, 95% CI: 0.55–3.01; other therapies HR: 0.54, 95% CI: 0.29–1.03; *p* < 0.01) [15]. A Markov model analysis supports these results, showing a slightly longer OS for levatinib after atezo/bev than for sorafenib after atezo/bev [51]. In the present study, patients who received subsequent therapy showed improved mOS. However, these results should be interpreted with caution due to the potential for immortal time bias, and they merely suggest the effectiveness of the “former” 2L medications in third or further line therapy.

The present study has some limitations. On the one hand, the retrospective study design introduces selection and underreporting bias (e.g., the simultaneous inclusion of 1L atezo/bev and lenvatinib), and on the other hand, the patient number for subgroup analyses is limited, so that its transferability to clinical decision-making is limited. No data on quality of life were available, and only adverse events of grade 3 or higher in a reduced patient number could be assessed. Moreover, although including lenvatinib-treated patients raises concerns about potential overlapping resistance mechanisms in a TKI-to-TKI progression, our data did not reveal significant differences in overall survival between patients treated with atezo/bev and those receiving lenvatinib in the first-line setting. Excluding these patients would have further reduced our cohort; hence, this remains a limitation that should be addressed in future studies. Nevertheless, to the best of our knowledge, SORA-2 represents the first multicentric European analysis of the effectiveness and tolerability of sorafenib in the 2L therapy of advanced HCC. Given the scarcity of evidence regarding the use of sorafenib in this setting, this study can significantly contribute to clarifying the recommendations for sorafenib use after ICI or TKI.

## 5. Conclusions

In summary, in a large unselected European cohort, sorafenib exhibited comparable efficacy in terms of PFS and ORR in 2L treatment as in first-line application and other established 2L therapy regimens. In 2L use, a slightly increased toxicity was observed compared to first-line treatment, and the median OS of the total cohort was somewhat shorter than with other approved 2L therapies. However, patients with preserved liver function (Child–Pugh class A) benefited significantly more, and with concomitantly low AFP levels < 400 ng/mL, patients achieved a median OS that was similar to the longest reported mOS from prospective first-line studies. In light of this, future studies should analyze the role of TKI in the sequential therapy of advanced HCC in a more differentiated manner based on specific patient characteristics.

## Published Materials

This article is a revised and expanded version of a paper entitled “Evaluating Sorafenib as Second-Line Treatment for Advanced Hepatocellular Carcinoma: SORA-2, a European Retrospective Multicenter Study”, which was presented at ESMO-GI, Munich on the 27th of June, 2024 [52].

## Figures and Tables

**Figure 1 cancers-17-00972-f001:**
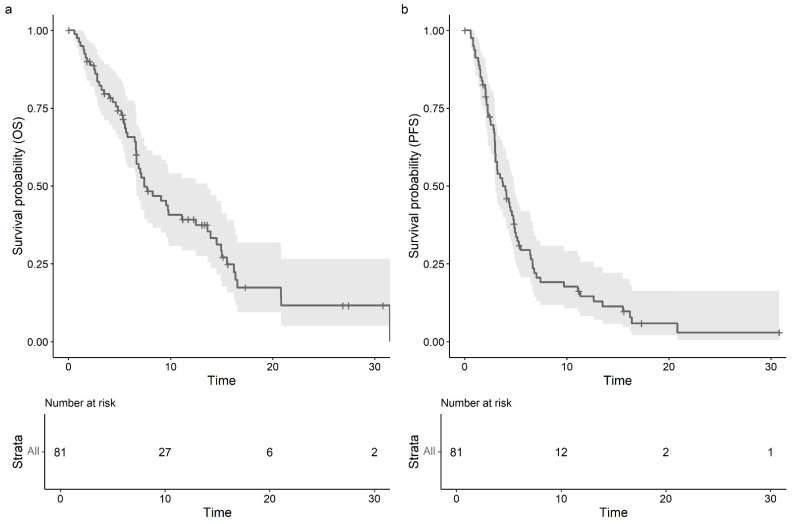
(**a**) Kaplan–Meier estimate for overall survival (OS) in months with median OS of 7.43 months (95% CI: 6.64–13.60). (**b**) Kaplan–Meier estimate for progression-free survival (PFS) in months with median 3.75 months (95% CI: 3.02–4.86).

**Figure 2 cancers-17-00972-f002:**
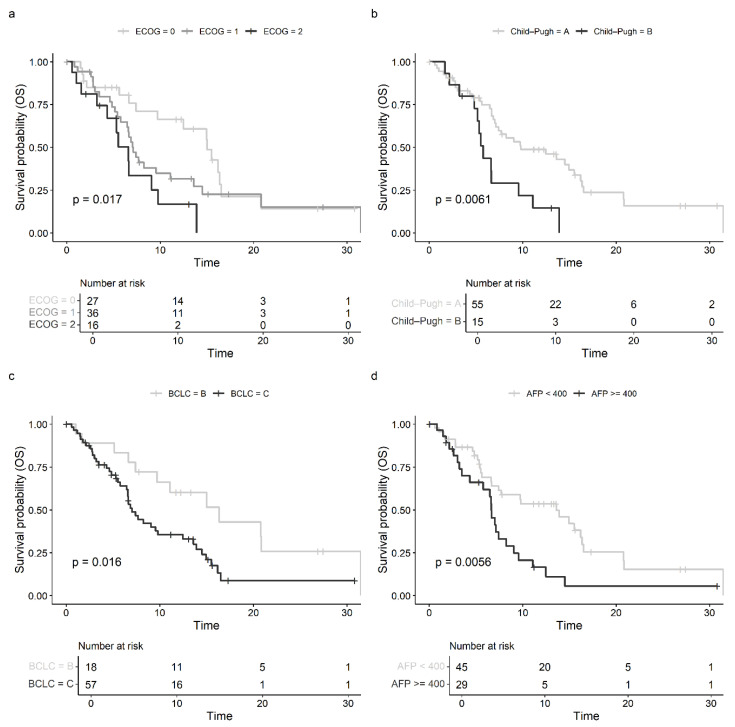
(**a**) Kaplan–Meier estimates with log-rank p for overall survival (OS) in months depending on ECOG status before sorafenib. (**b**) Kaplan–Meier estimates with log-rank p for overall survival (OS) in months depending on Child–Pugh class before sorafenib. (**c**) Kaplan–Meier estimates with log-rank p for overall survival (OS) in months depending on BCLC stage before sorafenib. (**d**) Kaplan–Meier estimates with log-rank p for overall survival (OS) in months depending on AFP level (<400 ng/mL vs. ≥400 ng/mL) before sorafenib.

**Figure 3 cancers-17-00972-f003:**
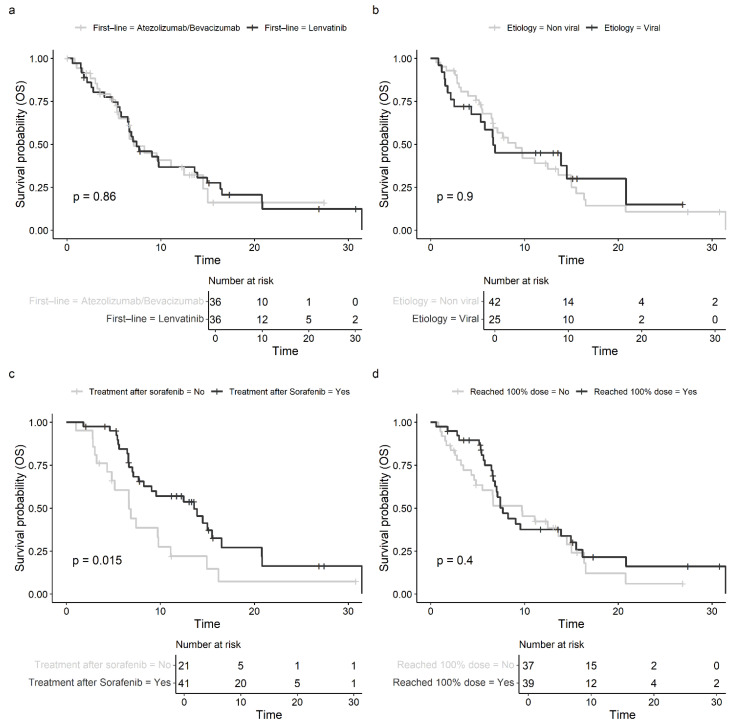
(**a**) Kaplan–Meier estimates with log-rank p for overall survival (OS) in months depending on first-line therapy regime. (**b**) Kaplan–Meier estimates with log-rank p for overall survival (OS) in months depending on etiology of cirrhosis (viral vs. non-viral). (**c**) Kaplan–Meier estimates with log-rank p for overall survival (OS) in months depending on further treatment status after sorafenib. (**d**) Kaplan–Meier estimates with log-rank p for overall survival (OS) in months depending on maximum sorafenib dose (100% reached yes vs. no).

**Figure 4 cancers-17-00972-f004:**
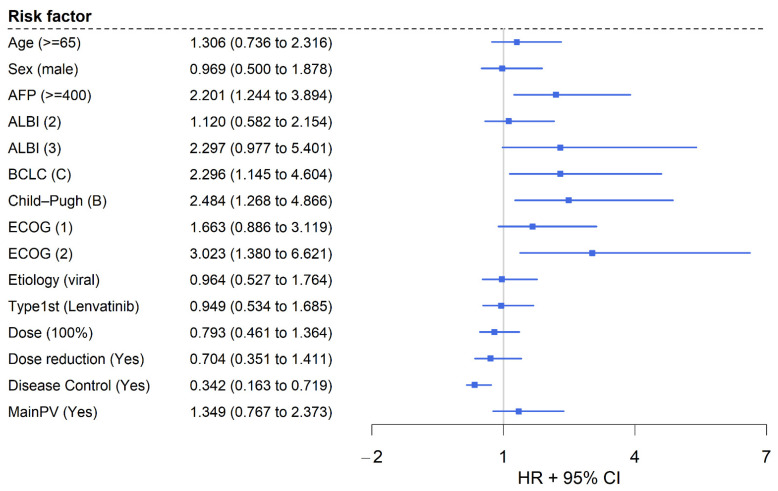
Forest plot for univariable analysis of overall survival depicting hazard ratios and corresponding 95% CIs. Reference values for HR of presented parameters are as follows: Age (<65), Sex (female), AFP (<400), ALBI (1), BCLC (B), Child–Pugh (A), ECOG (0), Etiology (non-viral), Type1st (atezolizumab/bevacizumab), Dose (100% not estimable), Dose reduction (No), Disease Control (No), MainPV (No). MainPV, main portal invasion; NLR, neutrophile–lymphocyte ratio.

**Figure 5 cancers-17-00972-f005:**
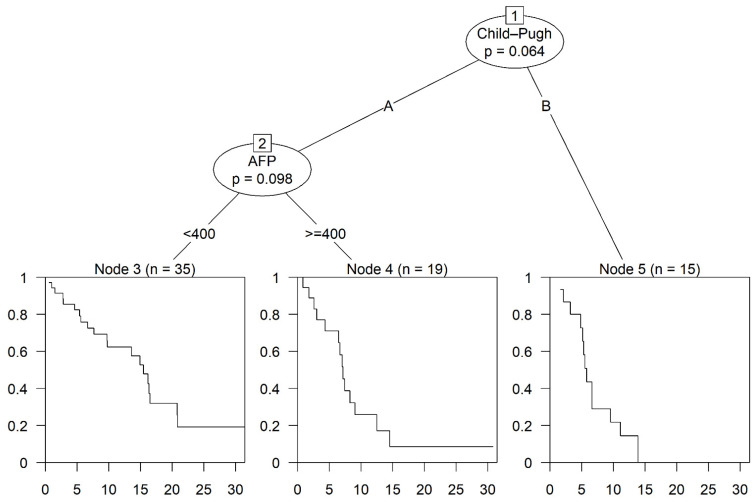
Multivariable analysis of overall survival based on survival tree with conditional inference permutation tests. Node 3 with median OS of 15.51 months (95% CI: 9.79–20.83). Node 4 with median OS of 7.13 months (95% CI: 6.47—not estimable). Node 5 with median OS of 5.78 months (95% CI: 5.16—not estimable).

**Figure 6 cancers-17-00972-f006:**
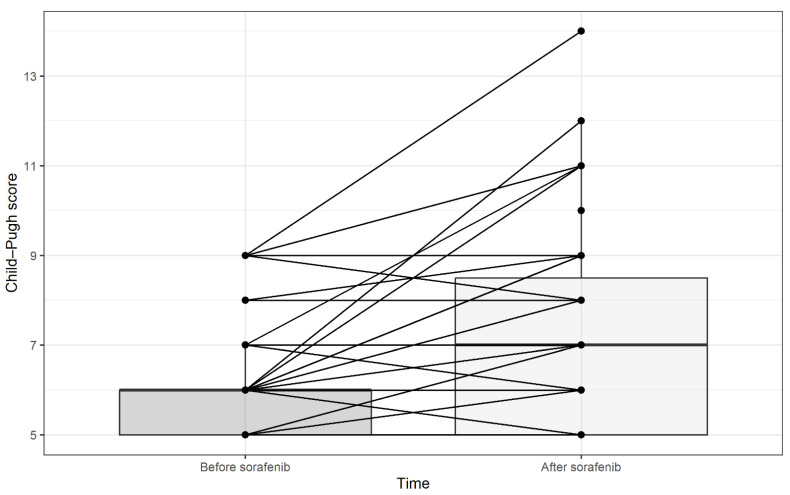
Paired line plot for Child–Pugh score before and after sorafenib treatment. Boxes represent median (bold line) with quartiles before and after sorafenib.

**Table 1 cancers-17-00972-t001:** Baseline characteristics.

	Overall (N = 81)
**Gender**	
Female	17 (21.0%)
Male	64 (79.0%)
**Age**	
Median [Min, Max]	67.8 [42.1, 83.7]
**ECOG status before sorafenib**	
0	27 (33.3%)
1	36 (44.4%)
2	16 (19.8%)
Missing	2 (2.5%)
**Etiology of cirrhosis**	
Non-viral	41 (50.6%)
Viral	25 (30.9%)
Other	1 (1.2%)
Missing	14 (17.3%)
**Non-viral**	
ASH	28 (34.6%)
MASLD/MASH	13 (16.0%)
**Viral**	
Hepatitis B	7 (8.6%)
Hepatitis C	18 (22.2%)
**Child–Pugh stage and score before sorafenib**	
**A**	55 (67.9%)
5	31 (38.3%)
6	24 (29.6%)
**B**	15 (18.5%)
7	8 (9.9%)
8	3 (3.7%)
9	4 (4.9%)
Missing	11 (13.6%)
**ALBI-Score before sorafenib**	
1	22 (27.2%)
2	39 (48.1%)
3	11 (13.6%)
Missing	9 (11.1%)
**BCLC before sorafenib**	
A	1 (1.2%)
B	18 (22.2%)
C	57 (70.4%)
D	1 (1.2%)
Missing	4 (4.9%)
**Macrovascular invasion before sorafenib**	44 (54.3%)
Thereof main portal vein invasion	25 (69.4%)
**Extrahepatic tumor spread**	47 (58.0%)
**AFP at before sorafenib**	
<400 ng/mL	45 (55.6%)
≥400 ng/mL	29 (35.8%)
Missing	7 (8.6%)
**Previous surgical and local therapy**	
TACE	35 (43.2%)
RFA	6 (7.4%)
MWA	11 (13.6%)
TARE	6 (7.4%)
Resection	27 (33.3%)
**Systemic first-line therapy**	
Atezolizumab/bevacizumab	35 (43.2%)
Lenvatinib	36 (44.4%)
Clinical trial with ICI	10 (12.3%)

ASH, alcoholic steatohepatitis; BCLC, Barcelona Clinic Liver Cancer; ECOG, Eastern Cooperative Oncology Group; ICI, immune checkpoint inhibitor; MASH, metabolic dysfunction-associated steatohepatitis; MASLD, metabolic dysfunction-associated steatotic liver disease; MWA, microwave ablation; RFA, radiofrequency ablation; TACE, transarterial chemoembolization; TARE, transarterial radioembolization.

**Table 2 cancers-17-00972-t002:** Therapy characteristics.

	Overall (N = 81)
**Patients reaching 100% sorafenib dose**	39 (48.1%)
**Patients requiring sorafenib dose reduction**	31 (38.3%)
**Reasons for sorafenib treatment cessation**	
Death	9 (11.1%)
Patient wish	2 (2.5%)
Progress	32 (39.5%)
Toxicity	23 (28.4%)
Missing	15 (18.5%)
**Treatment of HCC after sorafenib**	41 (50.6%)
**Number of subsequent treatments**	
1 subsequent treatment	38 (46.9%)
2 subsequent treatments	11 (26.8%)
3 subsequent treatments	4 (9.8%)
**Subsequent Treatments**	
Systemic third-line treatment	38 (46.9%)
Atezolizumab/bevacizumab	2 (5.0%)
Cabozantinib	26 (65.0%)
Lenvatinib	2 (5.0%)
Nivolumab/ipilimumab	2 (5.0%)
Ramucirumab	2 (5.0%)
Regorafenib	4 (10.0%)
MWA	1 (2.4%)
RFA	1 (2.4%)
Stereotactic body radiotherapy	1 (2.4%)
TARE	1 (2.4%)
Missing information about further treatment	2 (4.8%)
**Therapy response**	**N = 55**
Complete response (CR)	0 (0%)
Partial response (PR)	3 (5.5%)
Stable disease (SD)	20 (36.4%)
Progressive disease (PD)	32 (58.2%)
Objective response rate (ORR = CR + PR)	3 (5.5%)
Disease control rate (DCR = CR + PR + SD)	23 (41.8%)

MWA, microwave ablation; RFA, radiofrequency ablation; TARE, transarterial radioembolization.

**Table 3 cancers-17-00972-t003:** Incidence of grade ≥3 adverse events.

Adverse Event CTCAE Grade 3–5	N = 81
No	28 (34.6%)
Yes	41 (50.6%)
Missing	12 (14.8%)
**Adverse Events**	**N = 69**
Hand-foot skin reaction	10 (14.4%)
Liver dysfunction	9 (13.0%)
Diarrhea	8 (11.6%)
Fatigue	5 (7.2%)
Rash	4 (5.8%)
Nausea	2 (2.9%)
Arterial hypertension	2 (2.9%)
Renal insufficiency	2 (2.9%)
Cornea implant rejection	1 (1.4%)
Bleeding	1 (1.4%)
Mucositis	1 (1.4%)
Myocardial infarction	1 (1.4%)
Tremor	1 (1.4%)
Facial paresis	1 (1.4%)
Voice changes	1 (1.4%)

## Data Availability

Data are available in a publicly accessible repository at the following link: https://github.com/jmober (accessed on 10 March 2025).

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
