# Peer review of "Evaluating Sorafenib (SORA-2) as Second-Line Treatment for Unresectable Hepatocellular Carcinoma: A European Retrospective Multicenter Studyâ€"

_cancers, 2025, doi:10.3390/cancers17060972_

Round 1
Reviewer 1 Report
Comments and Suggestions for Authors
The manuscript “Evaluating Sorafenib as Second-Line Treatment for unresectable Hepatocellular Carcinoma: SORA-2, a European Retrospective Multicenter Study” by Möhring et al. presents a valuable study on the effectiveness of sorafenib as a second line (2L) treatment for unresectable hepatocellular carcinoma (HCC). This work provides crucial insights into clinical outcomes associated with sorafenib addressing the knowledge gap by evaluating the median overall survival, median progression-free survival, radiological therapy responses and toxicity profiles in a real-world multicenter European cohort.
At this stage, I did not identify any major concerns or critical issues within the manuscript. This study is a well-executed and a meaningful contribution to the evolving field of advanced HCC treatment. It is well-structured, methodologically sound, and clinically relevant. Overall, this work could serve as a significant reference for oncologist managing advanced HCC.
The following minor refinements could further strengthen the manuscript:
- Figures 1-3: The letter numbering is missing and should be revised to improve clarity and readability.
- Figure 4: A brief description of the results from the forest plot would enhance the interpretation of the findings.
- Figure 5 appears twice. Please revise the Child Pugh score figure to be labelled as Figure 6.
- Discussion: Are the authors aware of studies evaluating combination therapy with sorafenib as a second-line treatment? Could combination therapy lead to improved overall survival?
- Is there a difference known in toxicity profiles between sorafenib used as first-line versus a second-line treatment strategy? Is sorafenib more tolerable in the 2L context?
- The discussion could benefit from a clearer path for future investigations. Are there any potential biomarkers (genomic biomarkers, immune/inflammatory markers, miRNA) for treatment response that could be addressed in future?
Author Response
First of all, we would like to thank the reviewer for their constructive criticism. In the following, we will address the comments provided.
The following minor refinements could further strengthen the manuscript:
Comment: Figures 1-3: The letter numbering is missing and should be revised to improve clarity and readability.
Response: We completely agree with the reviewer; this improves the clarity of the figures and enhances overall readability. We have adjusted the figure labeling accordingly.
Comment: Figure 4: A brief description of the results from the forest plot would enhance the interpretation of the findings.
Response: We have added a brief explanatory legend to Figure 4, detailing the hazard ratio references and providing a clearer interpretation of the univariable analysis, as requested.
Comment: Figure 5 appears twice. Please revise the Child Pugh score figure to be labelled as Figure 6.
Response: Unfortunately, a minor oversight occurred on our part. We appreciate the reviewer’s careful revision and have corrected the error.
Comment: Discussion: Are the authors aware of studies evaluating combination therapy with sorafenib as a second-line treatment? Could combination therapy lead to improved overall survival?
Response: We are grateful for the reviewer’s important question regarding combination regimens involving sorafenib in advanced HCC. We have now incorporated a new paragraph in the Discussion summarizing some trials that explore sorafenib combined with immune checkpoint inhibitors. Preliminary evidence suggests that such combinations could indeed provide additional clinical benefit, although confirmatory data from larger randomized studies are still lacking..
Comment: Is there a difference known in toxicity profiles between sorafenib used as first-line versus a second-line treatment strategy? Is sorafenib more tolerable in the 2L context?
Response: We appreciate this important question. In the revised Discussion, we have expanded our comparison between toxicity outcomes in our 2L cohort and those reported in the first-line SHARP trial. Although no prospective trial has directly compared sorafenib toxicity in the first- versus second-line setting, our findings—as well as data from real-world studies (e.g., GIDEON)—suggest that patients with poorer baseline conditions and more advanced liver disease (e.g., Child-Pugh B), which is more typical of second-line patients, often exhibit a higher incidence of grade ≥3 adverse events. We have further clarified this reasoning in the manuscript, noting that our cohort’s elevated rates of hepatotoxicity, hand-foot syndrome, and other adverse events can be partly attributed to the poorer baseline condition of second-line patients. We would be grateful for any further suggestions if the reviewer feels additional clarification is needed.
Comment: The discussion could benefit from a clearer path for future investigations. Are there any potential biomarkers (genomic biomarkers, immune/inflammatory markers, miRNA) for treatment response that could be addressed in future?
Response: We fully agree with the reviewer that that exploring predictive biomarkers is crucialfor optimizing individualized therapy However, no single marker has yet been conclusively validated for routine clinical use. For example, while baseline AFP levels, early AFP changes, the neutrophil-to-lymphocyte ratio, and molecular alterations (e.g., CTNNB1 mutations, VEGF receptor expression) have shown some associations with response, their predictive value remains inconsistent. Recently, the CRAFITY score—which integrates baseline CRP and AFP levels—has demonstrated prognostic significance not only in patients receiving PD-(L)1-based immunotherapy but also in a sorafenib-treated cohort. We have now included this point in the revised manuscript and expanded our discussion on the need for further prospective validation of these biomarkers.
Reviewer 2 Report
Comments and Suggestions for Authors
Thank you for the opportunity to review the manuscript. The study identifies the challenge of the new area of 2L treatment sequencing following 1L ICI in HCC. The authors highlight that data supporting sorafenib in the 2L limited and the factors influencing outcomes after initial treatment failure are unclear. The manuscript is interesting but can be improved. There are additional retrospective studies of SOR in 2L after A/B that reduce study novelty, especially as related to the lack of data in European populations.
Questions/Comments
PFS 3.7mo. and OS 7.4mo., is this comparable to other 2L options after A/B for low CP, ECOG, AFP patients to other current options?
For Table 1, consider a different sublevel grouping strategy to improve the readability of the table. For instance, under cirrhosis etiology, first show the subgroup of viral / nonviral / missing, then in a separate viral subsection it is breakdown, then the non-viral group, and so forth. This would also apply to the Child-Pugh table (A v. B first, then numerically A5 – B9).
I am having trouble with the rationale to include Lenvatinib (LEN)1L in this study, where the primary focus is 2L outcomes with SOR following ICI. Although the precise mechanisms of resistance and mutational adaption after ICI or TKI are complex and continually developing, there are some overlapping mechanisms to adaption with LEN and SOR compared to ICI and SOR which may negatively influence patient outcomes in a TKI-to-TKI treatment progression compared to an ICI to TKI treatment progression. I realize excluding these patients may compromise the cohort size, especially when comparing to other 2L SOR studies that need to be added to the manuscript introduction / discussion. If a subgroup analysis is not ideal, this should at least be addressed as a potential limitation to the study.
There are many instances where sentences begin with an Arabic numeral that require revision.
When was radiological response assessed in relation to treatment initiation?
Regarding Figure 2, ln. 290, there is mention of selected subgroup significance (2 levels) between ECOG levels derived from the KM log-rank analysis which contains 3 levels. Can this intergroup comparison be obtained from the log-rank output? Figure 4 confirms this be an ECOG-2 v. ECOG-0 difference assuming the unlisted univariate factor (in this case, ECOG-0) is the reference for both comparisons. The methods and figure legend should be revised to clarify the reference group.
In Ln. 320, to clarify, this is comparing OS from SOR initiation stratifying based on 1L, not total OS from 1L to endpoint (including both 1L and 2L subgrouping by A/B-SOR vs. LEN-SOR)?
Regarding Figure 6 (the second of two figures labeled Figure 5), I understand what the authors are trying to show, but I think this might be better in a tabular format. This plot is treating the Child-Pugh score as continuous rather than categorical variable and the y-axis depicts tick values which do not correspond to actual CP scores. Also, presumably, the boxes (which are not described in the legend) correspond to a measure of central tendency, which also would not be appropriate for describing a CP distribution.
The authors should review the citations. For instance, Ln. 383 citation 34 points to a review article on treatment strategies involving TACE to support the statements in Ln 379-383, challenging the rationale for SOR after ICI potentially being beneficial through targeting different mechanisms. This review article does not support the author’s statement. The authors would need to cite the primary source of direct clinical evidence.
Ln.407 needs a revision to reflect very recently published studies. More generally, the paragraph beginning on Ln. 406 also needs a revision to reflect that current research along with the authors findings may establish SOR expectations in the 2L, but also set up the hypothesis that SOR may be inferior to LEN in the 2L.
Ln. 493 “The Asian heritage of these studies…” should be revised to identify the characteristics of the cohort affecting comparability more explicitly.
The 4pg. discussion is difficult to follow and reads as though there are a series of mini-review articles stacked together. I would recommend focusing the discussion only on those topics related or comparable to the author’s study design. The authors highlight and review clinical trials which are not related to their study design and results. The section(s) on ideal treatment responders (preserved liver function, excellent performance status, low-tier AFP) are all well-established prognostic factors and the authors correctly identify that outcome comparisons across different trials in well selected patients is complex.
Other Evidence of SOR in 2L
Ref. 22 [33977087] LEN PFS 6.1mo. SOR 2.5mo., OS 16.6mo vs. 11.2mo.
Ref. 30 PFS LEN 4.0mo. vs. 2.3mo. and OS (8.0 vs. 6.3 months) vs. SOR
Improvement in response rate but modest improvement in median PFS with LEN vs. SOR [4.9mo v 3.3mo] (not OS), even milder when matched (n = 123), following 1L A/B failure. (39875560) Korea Japan
Large study (n = 891) showed significant OS improvement with LEN v. SOR, 18.9mo. v 14.3mo. (39396495)
406 patient study, ECOG<2 and lack of VI associated with PFS, shorter OS for SOR 8.4mo. compared to 2L ICI (39877031) Italy
3.3mo PFS and 6.9 OS with 2L SOR after A/B. [n=213] (39168753) Euro Asia NA
2L LEN v. SOR, 3.1mo. v. 2.0mo. and 12.5mo. v. 7.8mo. OS [n=151] [P969 ESMO Congress 2024]
[n=126] PFS 3.5mo. v. 1.8mo LEN v. SOR and 10.3mo v. 7.5mo. OS. (38468561) Korea
Author Response
First and foremost, we would like to sincerely thank the reviewer for this exceptionally detailed review of our manuscript. The input is immensely helpful in improving the quality of our work. In the following, we will address the comments.
Comment 1: PFS 3.7mo. and OS 7.4mo., is this comparable to other 2L options after A/B for low CP, ECOG, AFP patients to other current options?
Response 1: The number of studies on systemic therapy after atezo/bev is limited, and the analysis of the specific subgroup of comparatively fit patients is not examined in as much detail. In our overall cohort, sorafenib yielded a median PFS of 3.7 months and OS of 7.4 months, whereas in our subgroup with preserved liver function (Child-Pugh A), low AFP (<400 ng/mL), and ECOG 0, median OS reached 15.5 months. Retrospective analyses—such as those by Yoo et al., Lee et al., and the ARTE study—have reported median OS ranging from 6.3 to 11.2 months after atezolizumab/bevacizumab. Moreover, other agents have also been evaluated in this setting; for instance, one retrospective analysis showed that lenvatinib produced significantly better PFS and a trend to improved OS compared to sorafenib. Importantly, our study included also patients who received lenvatinib as first-line therapy in addition to those treated with atezolizumab/bevacizumab, and our subgroup analyses did not show significant differences in overall survival based on the first-line regimen. We have expanded on these comparisons in the revised Discussion to provide a clearer context for our findings.
Comment 2: For Table 1, consider a different sublevel grouping strategy to improve the readability of the table. For instance, under cirrhosis etiology, first show the subgroup of viral / nonviral / missing, then in a separate viral subsection it is breakdown, then the non-viral group, and so forth. This would also apply to the Child-Pugh table (A v. B first, then numerically A5 – B9).
Response 2: We have adjusted the order of cirrhosis etiology in Table 1 1 by grouping patients into Viral, Non-viral, Other, and Missing categories, and then providing a detailed breakdown (Hepatitis B and C for Viral; ASH and MASLD/MASH for Non-viral). In addition, we have restructured the Child-Pugh data by first presenting CP A versus CP B (bolded for clarity) followed by their corresponding numerical scores. These modifications have been implemented in the revised Table 1.
Comment 3: I am having trouble with the rationale to include Lenvatinib (LEN)1L in this study, where the primary focus is 2L outcomes with SOR following ICI. Although the precise mechanisms of resistance and mutational adaption after ICI or TKI are complex and continually developing, there are some overlapping mechanisms to adaption with LEN and SOR compared to ICI and SOR which may negatively influence patient outcomes in a TKI-to-TKI treatment progression compared to an ICI to TKI treatment progression. I realize excluding these patients may compromise the cohort size, especially when comparing to other 2L SOR studies that need to be added to the manuscript introduction / discussion. If a subgroup analysis is not ideal, this should at least be addressed as a potential limitation to the study.
Response 3: We appreciate the reviewer for highlighting this important point. Although there are theoretical concerns about overlapping resistance mechanisms in a TKI-to-TKI progression (lenvatinib to sorafenib) compared to an ICI-to-TKI progression, our data did not show any significant impact of the first-line regimen on overall survival under sorafenib. Given that excluding these patients would compromise our cohort size,, we have now added this relevant limitation in the revised discussion in the limitations section.
Comment 4: There are many instances where sentences begin with an Arabic numeral that require revision.
Response 4: We revised all sentences that began with an Arabic numeral to ensure adherence to proper stylistic conventions, as recommended.
Comment 5: When was radiological response assessed in relation to treatment initiation?
Response 5: In our study, the first radiological assessment was scheduled at 12 weeks after the initiation of sorafenib therapy. Subsequent imaging evaluations were performed every 12 weeks as standard practice; however, if clinical or laboratory findings indicated potential disease progression, an earlier assessment was carried out to ensure timely intervention. We have now further clarified this point in the Methods section, as required.
Comment 6: Regarding Figure 2, ln. 290, there is mention of selected subgroup significance (2 levels) between ECOG levels derived from the KM log-rank analysis which contains 3 levels. Can this intergroup comparison be obtained from the log-rank output? Figure 4 confirms this be an ECOG-2 v. ECOG-0 difference assuming the unlisted univariate factor (in this case, ECOG-0) is the reference for both comparisons. The methods and figure legend should be revised to clarify the reference group.
Response 6: The reviewer has correctly interpreted our results here. Figure 2 presents the outcome of the log-rank test, while Figure 4 displays the hazard ratios (HR) from the univariable analysis, with ECOG (0) serving as the reference for both ECOG stages. We have adjusted the legend of Figure 4 accordingly.
Comment 7: In Ln. 320, to clarify, this is comparing OS from SOR initiation stratifying based on 1L, not total OS from 1L to endpoint (including both 1L and 2L subgrouping by A/B-SOR vs. LEN-SOR)?
Response 7: We have added "2L" to "OS" to avoid creating a misleading impression for the reader.
Comment 8: Regarding Figure 6 (the second of two figures labeled Figure 5), I understand what the authors are trying to show, but I think this might be better in a tabular format. This plot is treating the Child-Pugh score as continuous rather than categorical variable and the y-axis depicts tick values which do not correspond to actual CP scores. Also, presumably, the boxes (which are not described in the legend) correspond to a measure of central tendency, which also would not be appropriate for describing a CP distribution.
Response 8: We have corrected the labeling of Figure 6. Regarding the Child-Pugh score and its scaling, we have adjusted the y-axis to allow for a more meaningful interpretation. Accordingly, the figure has been replaced. Since we consider the Child-Pugh score to be an ordinal measure, we have used box plots to represent the median and quartiles. The corresponding explanation has now been added to the figure legend. We hope the reviewer agrees with this approach.
Comment 9: The authors should review the citations. For instance, Ln. 383 citation 34 points to a review article on treatment strategies involving TACE to support the statements in Ln 379-383, challenging the rationale for SOR after ICI potentially being beneficial through targeting different mechanisms. This review article does not support the author’s statement. The authors would need to cite the primary source of direct clinical evidence.
Response 9: The reviewer is completely right. Accordingly, we replaced the citation 34 with a primary source that directly supports our statement regarding the rationale for using sorafenib after ICI.
Comment 10: Ln.407 needs a revision to reflect very recently published studies. More generally, the paragraph beginning on Ln. 406 also needs a revision to reflect that current research along with the authors findings may establish SOR expectations in the 2L, but also set up the hypothesis that SOR may be inferior to LEN in the 2L.
Response 10: We appreciate this valuable suggestion. We have revised the paragraph (lines 406-407) in accordance with the reviewer's comments to incorporate recent evidence comparing sorafenib and lenvatinib in the second-line setting after atezolizumab/bevacizumab. We have also acknowledged in the Discussion that these data raise the possibility that sorafenib could be inferior to lenvatinib in specific patient subgroups. In the following paragraph, we address the newly included references. We have also partially revised the corresponding paragraph in the introduction.
Comment 11: Ln. 493 “The Asian heritage of these studies…” should be revised to identify the characteristics of the cohort affecting comparability more explicitly.
Response 11: We appreciate the reviewer’s comment. We have revised the text to replace the reference to ‘Asian heritage’ with a more specific description of the cohort differences, such as a higher prevalence of hepatitis B–related HCC, younger age distribution, and varying comorbidity profiles, which may limit direct comparability to our predominantly Western-based population.
Comment 12: The 4pg. discussion is difficult to follow and reads as though there are a series of mini-review articles stacked together. I would recommend focusing the discussion only on those topics related or comparable to the author’s study design. The authors highlight and review clinical trials which are not related to their study design and results. The section(s) on ideal treatment responders (preserved liver function, excellent performance status, low-tier AFP) are all well-established prognostic factors and the authors correctly identify that outcome comparisons across different trials in well selected patients is complex.
Response 12: We understand the reviewer's argument; however, our intention in the discussion section was both to compare our results with similar studies and provide the reader with a broader overview of the existing second line treatment options—especially for those less familiar with the evolving therapeutic landscape—and to position sorafenib’s role more clearly in 2L therapy, even if its use has become less frequently in the era of IO-based first-line therapy. We have incorporated the valuable input from the reviewers to sharpen the focus on how our findings align with those of comparable studies, and we hope that this approach balances contextual breadth with sufficient clarity.
Comment 13: Other Evidence of SOR in 2L: Ref. 22 [33977087] LEN PFS 6.1mo. SOR 2.5mo.; OS 16.6mo vs. 11.2mo.; Ref. 30 PFS LEN 4.0mo. vs. 2.3mo. and OS (8.0 vs. 6.3 months) vs. SOR, Improvement in response rate but modest improvement in median PFS with LEN vs. SOR [4.9mo v 3.3mo] (not OS), even milder when matched (n = 123), following 1L A/B failure. (39875560) Korea Japan; Large study (n = 891) showed significant OS improvement with LEN v. SOR, 18.9mo. v 14.3mo. (39396495); 406 patient study, ECOG<2 and lack of VI associated with PFS, shorter OS for SOR 8.4mo. compared to 2L ICI (39877031) Italy; 3.3mo PFS and 6.9 OS with 2L SOR after A/B. [n=213] (39168753) Euro Asia NA; 2L LEN v. SOR, 3.1mo. v. 2.0mo. and 12.5mo. v. 7.8mo. OS [n=151] [P969 ESMO Congress 2024]; [n=126] PFS 3.5mo. v. 1.8mo LEN v. SOR and 10.3mo v. 7.5mo. OS. (38468561) Korea
Response 13: We appreciate the thorough list of additional references comparing sorafenib and lenvatinib in the second-line setting. We included the majority of the suggested studies into our revised Discussion, ensuring that both the improvements in PFS and OS with lenvatinib—and the context in which sorafenib remains a viable option—are clearly addressed.
Reviewer 3 Report
Comments and Suggestions for Authors
Summary:
This is a retrospective multicenter study of sorafenib as second-line treatment for unresectable hepatocellular carcinoma. N=81 patients from 12 European centers who received sorafenib as second line treatment are included. They identify a group with the best median overall survival using multivariate analysis, and this group is defined by CP A and AFP levels below 400ng/mL They conclude that further research to identify predictive factors for response and survival in order to optimize treatment algorithms in advanced HCC are needed. Overall this is a strong paper and analysis that should be of interest to readers.
Comments
The introduction was very strong, but I wondered if the discussion section, which was very long, could be shortened somehow? If not, then it is okay as is.
The statistical methods section should include details on what methods were used to construct confidence intervals for median survival times. There are various methods but no standard procedures. Some methods are based on asymptotic approximations, and performance on this relatively small dataset may be an issue. (CI's for hazard ratios are standard and do not need to be explained.) At any rate, the main point is to clearly state the method used.
Related to the last comment, having "not reached" as the right endpoint listed in several places thoughout for median survivial CI's seems not right terminology. Maybe "not estimable" would be more correct.
While the tree analysis is interesting and valuable, some discussion of whether these groupings are strong enough to really impact treatment seemed needed. It seemed like due to the high risk across all groups, perhaps these are of limited value for treatment decisions, since patients should generally be treated aggressively?
Author Response
First of all, we would like to thank the reviewer for their constructive criticism. In the following, we will address the comments provided.
Comment 1: The introduction was very strong, but I wondered if the discussion section, which was very long, could be shortened somehow? If not, then it is okay as is.
Response 1: We understand the reviewer's argument; however, our intention in the discussion section was both to compare our results with similar studies and provide the reader with a broader overview of the existing second line treatment options—especially for those less familiar with the evolving therapeutic landscape—and to position sorafenib’s role more clearly in 2L therapy, even if its use has become less frequently in the era of IO-based first-line therapy. We have incorporated the valuable input from the reviewers to sharpen the focus on how our findings align with those of comparable studies, and we hope that this approach balances contextual breadth with sufficient clarity.
Comment 2: The statistical methods section should include details on what methods were used to construct confidence intervals for median survival times. There are various methods but no standard procedures. Some methods are based on asymptotic approximations, and performance on this relatively small dataset may be an issue. (CI's for hazard ratios are standard and do not need to be explained.) At any rate, the main point is to clearly state the method used.
Response 2: We appreciate the reviewer’s concern regarding the methods used to construct confidence intervals for median survival times. In the revised Methods section, we have specified that we used a normal approximation based on the log(survival) function from the Kaplan-Meier estimates. We acknowledge that our sample size is relatively small, which may affect the precision of these intervals. We have added a corresponding passage to the methods section.
Comment 3: Related to the last comment, having "not reached" as the right endpoint listed in several places thoughout for median survivial CI's seems not right terminology. Maybe "not estimable" would be more correct.
Response 3: We appreciate the reviewer’s suggestion, as it contributes to more precise wording in our manuscript. We have replaced ‘not reached’ with ‘not estimable‘ and adjusted the manuscript accordingly.
Comment 4: While the tree analysis is interesting and valuable, some discussion of whether these groupings are strong enough to really impact treatment seemed needed. It seemed like due to the high risk across all groups, perhaps these are of limited value for treatment decisions, since patients should generally be treated aggressively?
Response 4: We appreciate the reviewer’s insight regarding the clinical applicability of our survival tree findings. Indeed, all subgroups identified by the tree still represent a population at high risk for poor outcomes, and our results do not suggest withholding second-line therapy from any particular subgroup. Rather, we consider this an exploratory analysis that may help stratify prognosis and guide future prospective trials focused on refining treatment strategies. We have now added a short paragraph in the Discussion clarifying that the survival tree is hypothesis-generating and requires validation in larger cohorts before it can be used to make definitive clinical decisions.
Round 2
Reviewer 2 Report
Comments and Suggestions for Authors
The authors have addressed my concerns. I find the manuscript presentation to be improved and have no remaining concerns.